# Enacting Mana Māori Motuhake during COVID-19 in Aotearoa (New Zealand): “We Weren’t Waiting to Be Told What to Do”

**DOI:** 10.3390/ijerph20085581

**Published:** 2023-04-19

**Authors:** Lynne Russell, Michelle Levy, Elizabeth Barnao, Nora Parore, Kirsten Smiler, Amohia Boulton

**Affiliations:** 1Te Hikuwai Rangahau Hauora, Health Services Research Centre, Te Herenga Waka—Victoria University of Wellington, Wellington 4310, New Zealand; 2Whakauae Research Services Ltd., Whanganui 4500, New Zealand

**Keywords:** Indigenous health, Māori, collectivity, COVID-19, self-determination, community-based responses, mana motuhake, Kaupapa Māori

## Abstract

Māori, the Indigenous people of Aotearoa (New Zealand), were at the centre of their country’s internationally praised COVID-19 response. This paper, which presents the results of qualitative research conducted with 27 Māori health leaders exploring issues impacting the effective delivery of primary health care services to Māori, reports this response. Against a backdrop of dominant system services closing their doors or reducing capacity, iwi, hapū and rōpū Māori (‘tribal’ collectives and Māori groups) immediately collectivised, to deliver culturally embedded, comprehensive COVID-19 responses that served the entire community. The results show how the exceptional and unprecedented circumstances of COVID-19 provided a unique opportunity for iwi, hapū and rōpū Māori to authentically activate mana motuhake; self-determination and control over one’s destiny. Underpinned by foundational principles of transformative Kaupapa Māori theory, Māori-led COVID-19 responses tangibly demonstrated the outcomes able to be achieved for everyone in Aotearoa when the wider, dominant system was forced to step aside, to be replaced instead with self-determining, collective, Indigenous leadership.

## 1. Introduction

Globally, COVID-19 has magnified existing inequities, with the pandemic consistently being found to disproportionately impact populations who experience institutional and structural racism [1]. For Māori, the Indigenous people of Aotearoa (New Zealand), COVID-19 has existed within a broader context of intergenerational health inequity, and a demonstrated history of ineffective and inadequate health care responses [2,3]. That Māori would bear the disproportionate burden of COVID-19 was of major concern to Māori health experts from the outset [4,5,6].

COVID-19 was first identified in Aotearoa in February 2020. With cases progressively rising, the government introduced a four-level alert system on 21 March 2020. On 25 March 2020, all of Aotearoa moved to Alert Level 4 ‘lockdown’, which saw everyone except essential workers instructed to remain in their own homes. This first nationwide lockdown lasted for 33 days until 28 April 2020, at which time the country moved to the less restrictive Alert Level 3 [7,8]. Aotearoa entered a second period of Level 4 lockdown on 17 August 2021, when the highly contagious Delta variant was detected in Auckland. The nationwide lockdown remained until 31 August 2021. Following this, regions in Aotearoa occupied different Alert Levels until 2 December 2021. Auckland remained in Alert Level 4 lockdown until 21 September 2021, followed by Alert Level 3 until 2 December 2021 [9].

The success of the internationally praised COVID-19 response in Aotearoa was underpinned by Māori-led responses [10,11]. The devastation wrought upon Māori by the 1918 Spanish Influenza epidemic, still present in living memory, contributed to the collective view that not only would Māori communities likely be most impacted by COVID-19 [2,3], but the government was unlikely to adequately protect these communities [10]. Reflecting the power dynamics and intent of a system in which Indigenous worldview, knowledge and experiences are deliberately marginalised [12], dominant system services closed their doors or reduced capacity. Against this backdrop, iwi, hapū (‘tribal’ collectives) and other rōpū Māori (Māori groups), including ‘tribal’ authorities, Whānau Ora organisations (community-based services specifically delivering a culturally based, family-centred approach to wellbeing), pan-Māori organisations, and community service providers, immediately mobilised to occupy a central position in COVID-19 responses across Aotearoa. Directly tailored to the specific risks and needs of communities, the extensive scope of the Māori-led COVID-19 response more closely resembled that of local or central government, as opposed to a community or sector response [2,10,11,13].

With a focus on providing leadership to ensure communities were well supported, particularly during the initial periods of restriction [10,14], Māori-led COVID-19 responses included: pandemic co-ordination hubs; checkpoints and road blocks (aimed at preventing unnecessary travel across regions); 0800 freephone support; static and mobile testing and vaccination clinics; financial support; emergency housing, including for whānau (extended families) returning from overseas; service co-ordination; transport; whānau and kaumātua (respected elders) welfare checks and support; food, hygiene, and care packs; assistance to access prescription medications; data connectivity; culturally relevant messaging; and online educational content [2,13,14,15]. Māori businesses offered practical support; there were digital innovations across organisations; and whānau themselves widely shared resources to uplift and support holistic wellbeing (e.g., karakia (traditional ‘prayers’), cooking and physical exercises) [2,13,14,16].

Transformative Kaupapa Māori theory, which uses Māori theorising as a vehicle for conscientisation, resistance and reflective action [17], assists to better understand the wider significance of Māori-led COVID-19 responses in Aotearoa. Explicitly concerned with not only Indigenous control over one’s own wellbeing and priorities, but also the impact of unequal power structures that conceal, perpetuate and maintain inequity for Māori, Kaupapa Māori theory has provided the foundation for active disruption in Aotearoa for over the past 40 years, particularly in education and research [17,18]. 

Central to Kaupapa Māori theory is the concept of mana motuhake. Mana motuhake is defined as “separate identity, autonomy, self-government, self-determination, independence, sovereignty, authority—mana through self-determination and control over one’s own destiny” [19]. The relationship of mana motuhake to Indigenous wellbeing in Aotearoa is well-established. For example, the enshrining of mana motuhake across primary health care in Aotearoa is essential to the arguments presented in the Waitangi Tribunal *Health Services and Outcomes Kaupapa Inquiry* [20]. Likewise, supporting the premise that the solutions to health inequity lie within self-determined hapori Māori (Māori communities) themselves [21], the government-initiated Health and Disability System Review in 2020 concluded mana motuhake must be embedded within health system reform if persistent inequities for Māori in Aotearoa were to be addressed [22]. 

This paper reports on the results of research conducted with 27 Māori health leaders that explored issues impacting the effective delivery of primary health care services to Māori. The results demonstrate how the exceptional circumstances of the COVID-19 pandemic provided a unique opportunity for iwi, hapū and rōpū Māori to authentically activate principles of Kaupapa Māori theory. This, in turn, tangibly demonstrated the outcomes able to be achieved for Aotearoa as a whole, when the wider, dominant system was forced to step aside to be replaced instead with mana Motuhake—self-determining, collective, Indigenous leadership. 

## 2. Methods

*Rāranga Tāngata*, *Oranga Tāngata* is one of six projects in a five-year research programme ‘Enhancing Primary Health Care Services to Improve Health in Aotearoa’ (formally named *Primary Insights: Aotearoa/New Zealand’s Primary Health Care Study*) at Te Hikuwai Rangahau Hauora | Health Services Research Centre, Te Herenga Waka–Victoria University of Wellington. Utilising Kaupapa Māori theory, which prioritises Māori worldview, knowledge and analytical frames [23], the research was authoritatively guided by the professional, clinical and cultural expertise of kaumātua and a Rōpū Kaitiaki who brought a wealth of experience to the study. Their expertise included mātauranga (knowledge), tikanga (Māori custom) and te reo Māori (the Māori language); iwi/hapū development; hauora Māori (Māori health/wellbeing); health and social services; health promotion; policy development; funding and planning within the health sector; Māori methodologies; and Whānau Ora (a culturally grounded, holistic approach to improving the wellbeing of whānau as a group and addressing individual needs within the context). 

Published literature on Māori perspectives relating specifically to health and social services throughout the pandemic was limited at the commencement of this study. An initial literature review was, therefore, widened to focus on Indigenous experiences of health and social services during pandemics or other global crises. This was conducted through an Internship from Ngā Pae o Te Māramatanga Internship (Aotearoa’s Māori Centre of Research Excellence funded by the Tertiary Education Commission and hosted by The University of Auckland).

Data collection began with a wānanga (group discussion) with members of the Rōpū Kaitiaki, at which an adaptation of Q methodology, utilised in the wider research programme, was trialled. Participants were presented with a number of key Māori primary health care issues, each accompanied by four associated statements intended to generate discussion. This research method, used in social sciences to systematically study participants’ ‘subjectivity’ or viewpoints on an issue by having them rank and sort a series of statements, was, however, deemed unsuitable by the Rōpū Kaitiaki. A less burdensome approach was instead agreed to, based on a collectively developed set of guiding questions that sought to understand what oranga (wellbeing) looked like for Māori, what a Māori response to achieving oranga was, and what was required for services to operate effectively towards this goal. Further questioning investigated how Māori maintained oranga during the COVID-19 lockdown as individuals, and how their iwi, hapū, whānau and hapori (communities, including local services) did the same [24].

A total of 22 interviews were undertaken using this Kaupapa Māori theory-based approach to data collection, with 27 leaders in Māori health, including iwi health/social service/education providers; nationally prominent and recognised Māori primary health care experts; and Ministry of Health and District Health Board (responsible for providing or funding the provision of health services in 20 districts of Aotearoa prior to 2022) (DHB) senior management. All interviews were conducted by Māori interviewers and recorded with participants’ consent. Some were conducted in person, kanohi ki te kanohi (face-to-face), others over Zoom; the time spent varied. Irrespective of the method, all incorporated tikanga and te reo Māori in their process, which included understandings of Indigenous data sovereignty. This understanding ensured participants were offered the opportunity to review and amend the transcripts of their interviews and determine their identification alongside any quote/s used of theirs in reporting of study findings, which supported their continued ownership of their experiences and stories. At the conclusion of each interview, the researchers also undertook a reflexive exercise through the discussion of their interpretations of the insights shared. As Indigenous researchers, this process allowed for acknowledgment of relationship with participants, whilst ensuring subjectivity did not influence research outcomes.

Transcribed interviews were iteratively analysed, the preliminary data initially being coded in NVivo, a software tool that helps organise qualitative data, and the full dataset later being manually coded, using a systematic, thematic approach to aggregate and disaggregate the identified high-level and sub-themes. The time taken by the researchers to familiarise themselves with the data, generate codes, construct and review themes, was needed. Classifying, sorting and arranging data in this way ensured the rich insights shared, produced clearly articulated, defensible evidence [24].

This paper reports on the themes identified that were specifically relevant to iwi, hapū and rōpū Māori responses to the COVID-19 pandemic in Aotearoa.

## 3. Results

Overall, six key themes relating to the experiences of Māori-led COVID-19 responses were identified from the data:Immediate Indigenous MobilisationMagnitude of Indigenous ResponseCollectivising for a Common Kaupapa (cause)Culturally Embedded Indigenous ResponsesGrowing Confidence and CapacityThe Return to ‘Business as Usual’

These themes assist in better understanding how iwi, hapū and rōpū Māori activated mana motuhake. The following exemplar, extracted from the data, captures most of these key themes.
***We weren’t waiting to be told what to do****I went and had a look and tracked the epidemics around the world … the impact on our Māori communities was horrific … We said, “We need a response to Māori needs. We need marae [building complex on iwi land that includes a whare tipuna (‘tribal’ meeting house)] clinics. We need foodbanks. We need to be able to provide vaccinations”. We didn’t know what that looked like. We needed to have troops on the ground. We needed to have first attenders, nurses, community workers. We need to know “Where are all the homes where our kaumātua are who won’t get access?”**So, we planned as the iwi. In the room it was great. All the Māori providers came to the first meeting, along with the Hastings District Council. They said, “What can we do?” The DHB sent some people. We just came together as a collective and started to map it out. Then they got TPK [Te Puni Kōkiri | Ministry of Māori Development] to say, “Okay, what else do we need?” and they go, “Well, we need to have centralised this…” so they gave him an office. They said, “We need people in each community”, and community people put their hand up and said, “I’ll be the person in my community”.**They rang each of the maraes [sic]. We had our work vans, we had four vans from here that went out to the marae and picked up kai [food] and went and delivered it. We had a list of all the kaumātua flats. We went to all the kaumātua flats and dropped resources. The girls here at work, the midwives and the nurses made packs up. We had handwashing, towels, handtowels, face masks and soap. They were all in little packs. We made up about 150 packs and just delivered them out there, with a message: “What does COVID look like? If you get these symptoms, don’t go, ring up. Ring us and we’ll come first”.**The Māori response was there. We weren’t waiting for funding. We weren’t waiting to be told what to do. We were doing it because we knew it had to be done. (Jean Te Huia (Ngāti Kahungunu), CEO/Founder, Kahungunu Health Services, August 2021)**[By convention, the ‘tribal’ affiliations of named Māori in this paper are placed in parentheses after their name.]*

### 3.1. Immediate Indigenous Moblisation

Participants in this study emphasised how, even before lockdown restrictions were formally announced, iwi, hapū and rōpū Māori immediately acted to mobilise significant and comprehensive COVID-19 responses that served the entire community, including across provincial and rural sites.


*The Māori providers in particular were at the front line … they organised themselves very quickly and were smart about it … One of the things I loved was the activation of our iwi. Because even before we formally went into the lockdown, our iwi group in our rohe [region] … came together and they didn’t wait for the Crown. They just activated themselves. (Rōpū Kaitiaki Wānanga, November 2020) [In attendance: Kaumātua Bill Kaua, Whaea Moe Milne, Gabrielle Baker, Wheturangi Walsh-Tapiata and Raranga Tāngata, Oranga Tāngata research team members]*


It was commented that the urgency by iwi, hapū and rōpū Māori to act instantaneously was driven by the knowledge that government agencies and their systems were too slow and controlling to effectively respond to community needs, especially in relation to testing and vaccination processes.


*What’s the DHB’s response for Māori? Because we’re going to die from this COVID … And there wasn’t one. So, every single week, every Monday at 11 o’clock, we had a Zui [Zoom meeting] and we wrote the plan, or we made them write the plan. (*
*Emeritus Professor Khyla Russell (Kāi Tahu, Kāti Māmoe, Waitaha, Rapuwai), Rakatira; Dr Justine Camp (Kāi Tahu, Kāti Māmoe, Waitaha, Rapuwai), Associate Dean (Māori), Otago Medical School, University of Otago*
*, May 2021)*


In addition, primary health care agencies and other social service providers were observed as, for a range of reasons, either reducing their capacity or closing their doors altogether during critical periods of the pandemic.


*Places like the [named] foodbank closed down, and that’s because the people who run it were all volunteers and they were all 60, 70 and 80. (Tracey Wright-Tawha (Kāi Tahu, Kāti Māmoe, Waitaha, Te Āti Awa), CEO, Ngā Kete Mātauranga Pounamu Charitable Trust, December 2020)*



*Because we knew that basically General Practices were going to start arguing about budget and money and how many they were going to get … she said, “For God’s sake, we’ll just do it then”. …*
*(*
*Emeritus Professor Khyla Russell (Kāi Tahu, Kāti Māmoe, Waitaha, Rapuwai), Rakatira; Dr Justine Camp (Kāi Tahu, Kāti Māmoe, Waitaha, Rapuwai), Associate Dean (Māori), Otago Medical School, University of Otago*
*, May 2021)*


### 3.2. Magnitude of Indigenous Response

Described by participants in this research as a ‘well-oiled machine’, iwi, hapū and rōpū Māori activated substantial COVID-19 responses across urban, rural and remote communities. An in-depth understanding of diverse hapori Māori and their needs, including the varying levels of resources available to communities, both prior to and during lockdowns, was critical to mobilising responses specifically tailored to meeting areas of greatest need. These responses included strategy, planning, systems, testing, vaccination, helplines, kai, hygiene packs, medications, welfare checks and isolation support.


*Strategy planning, set up systems, checking on kaumātua, organising Zooms with all the key people. We received over 5,000 phone calls in those four weeks, and we made about that many as well; … we did about 1,200 food parcels; we set up a foodbank; we were doing COVID testing every day; … We received 20,000 hygiene products and broke them up and distributed them all to the marae, and then we did about two or three hundred whānau as well … We were dropping off food so whānau could isolate. We were making those wellbeing checks. We were dropping off medications to kaumātua who were isolated. It was working like a well-oiled machine. (Tracey Wright-Tawha (Kāi Tahu, Kāti Māmoe, Waitaha, Te Āti Awa), CEO, Ngā Kete Mātauranga Pounamu Charitable Trust, December 2020)*


Participants emphasised how, with face-to-face service delivery no longer possible, providers and organisations were required to rapidly develop entirely new and innovative ways of working, often aided by technology.


*We got a phone call that afternoon saying, “Guess what? Hurry up, you’ve got to set up a kind of testing clinic within 24 h” … We all ran in on the Sunday, did all the modelling, all the funding, and had it set up by Monday morning … two days later … we had to set up a community-based assessment centre … we got that up all running within a week… (*
*Dr Chris Tooley (Ngāti Kahungunu), CE, Te Puna Ora o Mataatua, March 2021)*


Focused on operationalising fast, agile, safe and flexible responses as each subsequent wave of infection hit, iwi, hapū and rōpū Māori enhanced, streamlined and strengthened their responses throughout the pandemic. Responses were carefully analysed and adapted to best meet the needs of specific communities (e.g., taiohi, koroua and kuia (youth and elderly), those who were homeless, and those in rural areas).


*After about a week we noticed that no-one from the rural townships were coming into Whakatane [a town in the North Island of Aotearoa]. And why should they … the whole model that everyone should come to us. And so, we started designing the mobile units … had our mobile unit up and running during the third week of lockdown. (*
*Chris Tooley (Ngāti Kahungunu), CE, Te Puna Ora o Mataatua, March 2021)*


Distributing food and other whānau essentials (e.g., baby and childcare packages), alongside access to testing and vaccination were identified as priority pandemic response areas. However, other needs were also recognised as important during periods of restriction (e.g., connectedness for older people, addressing family harm and mental health support). Participants described the proactive ways iwi, hapū and rōpū Māori worked to ensure holistic community wellbeing.


*For kaumātua … we had to put in place things like setting up computers and the internet in people’s home and teaching them how to do Zoom. So, they could stay in touch with whānau. (Associate Prof Matire Harwood (Ngāpuhi), Kaupapa Māori Primary Care Physician, March 2021)*



*We went back to 2018 and all our people that had been on the non-violence program–e—to make sure they were okay, got support around them as well … there was about 170 I think we rang, and we rang every week … Ringing all the supplementary community housing places and making sure all the people with mental health illness were being seen by doctors or getting access into secondary care if they needed it … (Tracey Wright-Tawha (Kāi Tahu, Kāti Māmoe, Waitaha, Te Āti Awa), CEO, Ngā Kete Mātauranga Pounamu Charitable Trust, December 2020)*


Integral to the magnitude of pandemic responses activated by iwi, hapū and rōpū Māori was a dedicated focus on workforce development. Again, linked to need for instant responses, there was an intentional focus on upskilling existing community workforces to contribute across the range of essential areas (e.g., testing and vaccination). Whilst meeting an urgent need, the opportunity to upskill was also seen as presenting an opportunity to grow long-term workforce capacity that would be beneficial to communities well beyond the pandemic.


*We trained all their staff when we went around. So, their own nurses, and their own healthcare assistants … So, if and when they wanted to open their own mobile pop-up station, that they could do it straight away. (*
*Dr Chris Tooley (Ngāti Kahungunu), CE, Te Puna Ora o Mataatua, March 2021)*


### 3.3. Collectivising for a Common Kaupapa

The immediacy and magnitude of the response executed by iwi, hapū and rōpū Māori was possible because acting as a collective is a foundational principle in Te Ao Māori (the Māori world). Participants highlighted how iwi, hapū and rōpū Māori did not wait for directives from the government and their agencies regarding what they would be permitted to do. Importantly, they also did not wait for the provision of government resources prior to acting. Instead, they immediately collectivised; coming together to form alliances, which enabled the delivery of holistic, whānau-centred responses, to be instantly activated, in what was an unpredictable and rapidly changing environment.


*Everybody put in some pūtea [money]. We got a building from the council, and we stood up a kai distribution hub again. And we used our own money and some of our own networks to get the kai that was required over that period of time. (Materoa Mar (Ngāti Porou, Ngāti Whātua, Ngāpuhi), Upoko Whakarae: Te Tihi o Ruahine Whānau Ora Alliance, October 2021)*


This collectivising and sharing of resources also enabled iwi, hapū and rōpū Māori to enter into ‘high trust’ relationships with government agencies; that is, delivering services on the promise of future payment.


*During lockdown we were told to do something, and we never got funded until a couple of months later. It was based on high trust because you had to get things done straight away. We would be calling into the Ministry, just to touch base in relation to what we’re doing, and they’d be screaming down the phone saying, “Hurry up and get it done, we’ll sort it out later”. Which was a bit of a risk for us because no contract was signed. But we knew it had to be done. (*
*Dr Chris Tooley (Ngāti Kahungunu), CE, Te Puna Ora o Mataatua, March 2021)*


In some cases, new collectives and alliances, including pan-‘tribal’, quickly emerged to support coordinated and cohesive COVID-19 responses. Bridging across geographical regions, alliances and collectives provided deep reach within communities.


*If we look at whānau during this recent lockdown—the number of our whānau that are 70+ but are isolated. Being able to reach into those whānau because we’ve been able to join up our conversations to know, actually this person’s there. There’s a greater visibility around health, social, welfare. It’s being driven through iwi collaboration. (Hayden Wano (Taranaki, Te Āti Awa, Ngāti Tama, Ngāti Awa), CE, Tui Ora, September 2021)*



*We think we know our community quite well, but we were hearing from people that we’d never seen before … Some of them, they were just ringing up to say, “I’m from Bluff and I saw this elderly guy trying to carry his shopping from the Four Square” and I said, “Did you watch him go to his address?” And they said, “Yeah …“ … So, we would just go down and take supplies and knock and say, “You all right koro [affectionate term of address to an elderly male]*
*? Can we get it for you next time around?” (Tracey Wright-Tawha (Kāi Tahu, Kāti Māmoe, Waitaha, Te Āti Awa), CEO, Ngā Kete Mātauranga Pounamu Charitable Trust, December 2020)*


Importantly, with iwi, hapū, and rōpū Māori initiating and driving COVID-19 pandemic responses, relationships, including those with government agencies, became redefined, particularly in the early days of the pandemic.


*… within four or five days we were collectively talking to one another. Who’s doing what, when, how, why? … The Police were working with you. Medical staff were working with you. Civil Defence were working with you. Foodbanks were ringing … it was great for redefining relationships. (Tracey Wright-Tawha (Kāi Tahu, Kāti Māmoe, Waitaha, Te Āti Awa), CEO, Ngā Kete Mātauranga Pounamu Charitable Trust, December 2020)*



*On the roadblocks, it had the iwi groups, you had the DHB groups, and we had the Police and everybody else. And so, what was really important about that was the iwi had a big say in what happened …*
*(Rōpū Kaitiaki Wānanga, November 2020)*



*Every day at two o’clock we would have a Zoom with iwi Māori leaders across the rohe … the Māori staff who were sort of in senior roles inside of agencies. So, that worked very well for bringing together the resources that we needed for the connectivity; for what it was that we were aspiring to do, and how can we work together to make that happen. (Materoa Mar (Ngāti Porou, Ngāti Whātua, Ngāpuhi), Upoko Whakarae: Te Tihi o Ruahine Whānau Ora Alliance, October 2021)*


### 3.4. Culturally-Embedded Indigenous Responses

Participants emphasised how iwi, hapū and Māori organisations had from the outset positioned their COVID-19 responses within a value-driven cultural context. That responses were visibly situated within the culturally located concepts of whakapapa (ancestry), tikanga, whānau and manaakitanga (care for others) was seen as fundamental to the success of Indigenous-led pandemic responses in Aotearoa.


*In fact, those who actually reframe the whole COVID kaupapa (issue) in terms of our tikanga and our whakapapa … those are the ones that are having the greatest success … (Helmut Modlik (Ngāti Toa Rangatira), CEO, Te Rūnanga o Toa Rangatira, September 2021)*



*Running competitions … you know, hand-sanitising or washing your hands. Just trying to lighten some of that mood, but also reminding ourselves about what are the things that will protect our whakapapa.*
*(Materoa Mar (Ngāti Porou, Ngāti Whātua, Ngāpuhi), Upoko Whakarae: Te Tihi o Ruahine Whānau Ora Alliance, October 2021)*


Participants also commented how iwi, hapū and rōpū Māori had prioritised building trusting relationships as the foundation for community engagement and participation. Central, therefore, to the culturally embedded responses of iwi, hapū and rōpū Māori was a workforce who were able to facilitate trusting relationships across multiple levels, and ensuring the right people and processes were in place.


*The thing that you’re selling is trust. That’s what people need. You’re brokering trust, and you’re brokering those relationships and you’re brokering engagement. That’s the thing that you’ve got to secure first, and then you can talk about whatever services they need after that … (*
*Dr Chris Tooley (Ngāti Kahungunu), CE, Te Puna Ora o Mataatua, March 2021)*



*We were whānau-orientated. We knew our maraes [sic]. We knew our systems … there was a total connect with our whānau and the marae and things like that. We can do that again. (Jean Te Huia (Ngāti Kahungunu), CEO/Founder, Kahungunu Health Services, August 2021)*


Participants also emphasised how explicitly prioritising cultural concepts and values had resulted in a uniquely distinctive interaction for not only whānau, but service providers are well. For example, when guided by values such as whakapapa and manaakitanga, significant sacrifices were made by kaimahi (worker) to ensure others would not be placed at risk.


*Sometimes just, you know, a Māori kaimahi rocking up into their Pākehā [Caucasian] window, saying, “You all good? You need some kai? You need some prescriptions? How’s everything at home?” Some of them found it quite intrusive, it be like, “This is our business!” Whereas, all the Māori whānau, they rocked up, and as soon as they saw us, just opened the boot straight away, and all the boxes got piled in. Such a different understanding of our manaakitanga eh. (*
*Dr Chris Tooley (Ngāti Kahungunu), CE, Te Puna Ora o Mataatua, March 2021)*



*… the kaimahi didn’t go home because they’re not gonna take COVID back to their families … they didn’t wanna put their whānau at risk … Some of them were actually away from home for more than two weeks in a row …*
*(Rōpū Kaitiaki Wānanga, November 2020)*


The importance of iwi, hapū and rōpū Māori workforces having the freedom to prioritise and elevate cultural values when operationalising their COVID-19 responses was commented on. Related to this, some observed how there was no shortage of people wishing to help their own iwi and hapū pandemic responses, particularly in the rural areas.


*Our Māori nurses, primary healthcare … the freedom to be able to be in charge of their own domain … they’re out there doing TikTok; people are delivering them food; and they’re just getting through the numbers …*
*(Kerri Nuku (Ng*
*ā*
*ti Kahungunu, Ngāi Tai), Kaiwhakahaere, New Zealand Nurses Organisation, September 2021)*



*… we had people from all around the country who had whakapapa to each of those townships, who had a health background, all emailing us saying, “Hey, I want to be a part of this, and I want to be on that mobile unit, and help”. We had no shortage of people coming from around the country … be able to go back and do something for their own township. (*
*Dr Chris Tooley (Ngāti Kahungunu), CE, Te Puna Ora o Mataatua, March 2021)*


### 3.5. Growing Confidence and Capacity

Participants commented on the new level of confidence that had been developed and instilled across iwi, hapū and rōpū Māori as a result of their experiences responding to the COVID-19 pandemic. Facilitated in part by the collective mobilisation that had occurred, examples of ways in which confidence had grown included understanding how to utilise data to develop and inform pathways forward. With a long-term view extending beyond COVID-19, iwi, hapū and rōpū Māori were quick to implement collective learnings as the pandemic continued, meaning their responses became increasingly efficient and effective.


*There is a lot of work that’s been done around sovereignty and sharing information and that’s given an appetite for seeing data being critical to the way that iwi think and move … There’s a thirst for it, to understand more. That’s just been there before, but never in such a joined-up way.*
*(Hayden Wano (Taranaki, Te Āti Awa, Ngāti Tama, Ngāti Awa), CE, Tui Ora, September 2021)*


Also integral to that long-term view was fully embracing the opportunities COVID-19 presented as iwi, hapū and rōpū Māori were encouraged to extend beyond the limitations of their existing contracts or organisations. Some even documented their COVID-19 responses in innovative ways such as documentaries.


*Don’t be restricted by your organisation or your contract … If we’re kaupapa driven, figure out how it can happen. I found it quite an exciting time because those are some of the things that I think are fundamental to how I would like to practice. (Rōpū Kaitiaki Wānanga, November 2020)*



*We made a documentary called ‘Ka Puta Ka Ora’. It’s available on Māori TV On Demand and that gives a really good overview of the iwi Māori network and what we experienced, what whānau experienced, there’s some whānau who talk on there. And then, what were the things we learnt and what does that look like going forward? (Materoa Mar (Ngāti Porou, Ngāti Whātua, Ngāpuhi), Upoko Whakarae: Te Tihi o Ruahine Whānau Ora Alliance, October 2021)*


Participants observed how there was no going back from the strong iwi, hapū and rōpū Māori leadership asserted during COVID-19. Alongside this, iwi leaders themselves were seen as having gained an enhanced appreciation and understanding of the leadership and work their service providers undertook within the hauora (health/wellbeing) space.


*Trying to even to get to a hui to have those conversations with them [iwi] before were near on difficult, because they thought of you as those ‘health people’. Now, they understand. I think there’s a much better synergy. (Rōpū Kaitiaki Wānanga, November 2020)*


Whānau experiences of the services delivered during COVID-19 were also seen as having created new expectations in relation to service delivery for Māori communities. These expectations were seen as creating a form of leverage, which could be built on in the future.


*… we had created this new expectation around engagement … from a kind of mobile rural outreach point of view because that’s just the way they [whānau] want it. They don’t want to go to their doctor, for whatever reason, or their GP [General Practitioner] … we’re meeting that expectations [sic] of our whānau … all our whānau, who had never seen a doctor ever before, were coming out … (*
*Dr Chris Tooley (Ngāti Kahungunu), CE, Te Puna Ora o Mataatua, March 2021)*



*We want to be really forward-thinking and enabling, so people have lots of different ways that they can actually access service. I’m amazed at how many people just like being able to phone-in or Zoom-in, talk to their GP. It only takes 10–15 min, and then they can have their medication sent to them. They don’t have to come and get it. We make it easy. (Tracey Wright-Tawha (Kāi Tahu, Kāti Māmoe, Waitaha, Te Āti Awa), CEO, Ngā Kete Mātauranga Pounamu Charitable Trust, December 2020)*


### 3.6. The Return to ‘Business as Usual’

Participants emphasised how relationships common at the beginning of the pandemic, which had centered on the devolution of power to Māori, gradually started to disappear as a more centralised government agency approach re-emerged and COVID-19 issues began to be more overtly politicised. With central government agencies reverting to their ‘business as usual’ practices, some participants referred to the contrast as like being in ‘two different worlds’.


*… in the interim, for a very short time, they [government] became that back-office support. Until they were brought up to speed, and then they took the control back.*
*(Heather Skipworth (Ngāti Kahungunu, Ngāi Tahu, Ngāti Rangitihi, Ngāti Ruanui, Tūwharetoa), CEO IronMāori, September 2021)*



*What developed between this organisation, iwi down here, was almost like a trust relationship with the Crown agencies. But within three weeks of the lockdown being lifted, all I’m hearing all over the country, “They’re going right back to the old way”. All the trust stuff has disappeared out the door. (Rakatira (Kāi Tahu), March 2021)*



*Recovery post-COVID is a completely different story, where we appear to have gone back into same old, same old … we get invited to the table, but we get hōhā [annoyed] at the table, and we withdraw ourselves … it’s two different worlds … they’re pulling all the power back; they’re pulling it bit by bit.*
*(Rōpū Kaitiaki Wānanga, November 2020)*


It was commented that not only were Māori voices diminished as central government agencies resumed control, but non-Indigenous organisations also disturbingly started to convey the message that they were able to achieve the same outcomes as had iwi, hapū, rōpū Māori and hapori Māori. Participants described how, despite iwi, hapū and rōpū Māori being publicly praised for their COVID-19 responses, decision-makers continued to revert to a ‘one-sized-fits-all’ approach, even with a wealth of evidence demonstrating its ineffectiveness. Frustration was expressed that lessons from the many previous crises in which Māori had led or contributed significantly, were still not being learnt from.


*What we also noticed was that the Crown sort of started behaving like they could do the job that we had done previously … they didn’t peel away and say, “Here, get this resource out the door because you have got that covered”. They suddenly wanted to become us. (Materoa Mar (Ngāti Porou, Ngāti Whātua, Ngāpuhi), Upoko Whakarae: Te Tihi o Ruahine Whānau Ora Alliance, October 2021)*



*So, after the last lockdown, how the Prime Minister made the statement how amazingly iwi had stepped up and what an amazing job. When I got asked about it, I said, “Look, we hear that comment after every bloody disaster, but three months after that it’s completely forgotten”. And the next disaster, “Look how well iwi have stepped up”. We get sick of hearing that crap. (Rakatira (Kāi Tahu), March 2021)*


Reflective of this situation, participants described how the overall narrative surrounding Māori-led responses to COVID-19 also started to change. For some, this change in narrative was an attempt to conceal the inadequacies of the Crown-led response to COVID-19.


*When we came down to Level One, all of a sudden, “Oh, we’re vigilantes” … What they want to do is hide the racism stuff; is hide their inefficiencies and try and make this a Māori problem … (Rōpū Kaitiaki Wānanga, November 2020)*


Importantly, respondents observed that as the pandemic started to move out of the initial crisis period, iwi, hapū and rōpū Māori were not supported in continuing to build ongoing understanding and trust. Where power and decision-making had returned to a tightly centralised process controlled by the Ministry of Health, for example in relation to testing and vaccination policy and process, it was commented that improved outcomes would have unquestionably resulted, had Māori been in control.


*A lot of our people are deeply negative around the vaccine, and they run for disinformation really, really quickly … They may not be rabid anti-vaxxers, but they’re deeply concerned by the wider agenda of the New Zealand government, and they ought to be. (Simon Royal (Ngāti Raukawa, Ngāpuhi, Ngāti Whanaunga), CEO, National Hauora Coalition*
*, March 2021)*



*They’d fallen down so many bloody conspiracy rabbit holes that by the time the government started their campaign it was slow … Which meant all the anti-vax and all the conspiracy theories got in there. (Emeritus Professor Khyla Russell (Kāi Tahu, Kāti Māmoe, Waitaha, Rapuwai), Rakatira; Dr Justine Camp (Kāi Tahu, Kāti Māmoe, Waitaha, Rapuwai), Associate Dean (Māori), Otago Medical School, University of Otago*
*, May 2021)*


## 4. Discussion

Participants in this research highlighted the speed with which iwi, hapū and rōpū Māori responded to COVID-19, as well as the magnitude of those responses. As was identified in this and previous studies, not only were these responses instant, substantial and comprehensive, they were also adaptable, agile and flexible, as innovative and often entirely new ways of working with communities needed to be swiftly developed [2,13]. Strategies included existing funding and resourcing being quickly redeployed to support those most at risk, and existing workforces upskilled to cover a larger range of tasks [14]. Totally dispelling the perception that Māori lack agency in relation to their own health and wellbeing [10,21], as COVID-19 developments moved at an astonishingly rapid pace, iwi, hapū and rōpū Māori continuously analysed, enhanced, streamlined and strengthened their responses throughout the pandemic [2]. However, of importance is that Māori-led COVID-19 responses did not exist in a vacuum. Situated within a broader context of Indigenous aspirations for mana motuhake, iwi, hapū and rōpū Māori COVID-19 responses activated fundamental principles of Kaupapa Māori theory.

### 4.1. Culturally-Embedded Responses

Integral to Kaupapa Māori theory is the principle of validating and legitimating cultural aspirations and identity. This principle asserts and supports the centrality of te reo Māori, tikanga and mātauranga Māori (Māori ancestral knowledge). To ‘be Māori’, as encompassed in practices, values, and beliefs underpinned by Indigenous knowledge, is the taken-for-granted norm [25]. With the strength and resilience of Indigenous peoples at the forefront, culturally derived knowledge, analyses and processes influence whole systems, processes and outcomes, as opposed to isolated elements [18]. Consistent with previous research see [13,15,16], participants in this study directly attributed the success of Māori-led COVID-19 responses to these being explicitly embedded within shared and commonly understood cultural values such as whakapapa, whanaungatanga (relationships), manaakitanga and kaitiakitanga (guardianship).

With shared understandings of these everyday cultural values providing the foundation, iwi, hapū and rōpū Māori immediately comprehended the need for urgency in responding to COVID-19. For example, the fundamental importance of protecting and preserving whakapapa underpinned the urgency of responses [2,15]. Kaitiakitanga and manaakitanga responsibilities, which activate shared values of responsibility and duty to the wider community, are also intrinsically linked to whakapapa [11]. An holistic and relational cultural worldview, which unambiguously prioritised collective wellbeing, meant Māori-led responses were values-based and inclusive. Recognising COVID-19 lockdowns would present a range of issues for whānau, Māori-led responses extended well beyond a narrow focus on physical health [11,13,16].

This same value base was central to the COVID-19 Māori workforce. Tangibly effecting these cultural values when working with communities was essential, with the activation of everyday relational cultural values providing a solid foundation on which to build the trusting relationships necessary for quality engagement and positive outcomes [10,16]. Culturally-embedded responses from iwi, hapū and rōpū Māori as opposed to government agencies, were more likely seen as trusted sources of information by communities [2,15]. This was particularly important given that the deep distrust of government agencies had been heightened by the pandemic [16].

A COVID-19 Māori workforce was also assembled utilising shared cultural values of whakapapa, whanaungatanga, manaakitanga and kaitiakitanga. Supporting the findings of this research, previous studies describe how the essential work undertaken by paid and unpaid Māori workers who had quickly moved to fill substantial workforce gaps occurred within the context of “mahi aroha”—”work undertaken out of a love for people” [14] (p. 4). That there was no shortage of people wishing to assist their own iwi and hapū, particularly in rural and remote areas, illustrates the power these cultural values hold in mobilising whānau to act [16].

### 4.2. Collectivising for Action

Kaupapa Māori theory prioritises commitment by Māori communities to a collectively shared vision for political, social, economic and cultural wellbeing [26]. These shared visions serve to articulate and connect with Māori aspirations, thus providing powerful pathways to buy-in from Māori communities [17,18]. As was seen in this study, and consistent with what has been previously found see [2], integral to the immediate activation of COVID-19 responses by iwi, hapū and rōpū Māori was their ability to draw on a collective base of substantial skills, expertise and resources, including social infrastructure such as marae, strong governance systems, and strong connections and distributive relationships [5,10,11,13,16]. Of critical importance was that iwi, hapū and rōpū Māori did not hesitate, nor wait, for the provision of government resources. In the absence of clear direction and resourcing from government agencies, collectives quickly mobilised around shared visions and values. The approach premised upon collective action and the sharing of resources enabled deep reach into communities as Māori-led COVID-19 responses become primary distribution channels for government agencies [10].

This resulted in an astonishing amount of support being distributed to entire communities, not just hapori Māori, during the lockdown periods [2,10,13,16]. With Māori leadership driving responses, relationships with government agencies became significantly redefined. A long-term perspective, which extended beyond the pandemic, also saw Indigenous leadership prioritise ongoing data collection as part of COVID-19 responses. Increased understanding regarding how to utilise data to develop and inform pathways forward, not only increased confidence, but brought “a depth of knowledge about its many hapori Māori that was unparalleled, and certainly could not be emulated by Crown organisations” [2] (p. 10). With iwi, hapū and rōpū Māori quick to implement collective learnings as the pandemic continued, Māori-led COVID-19 responses became increasingly efficient and effective.

### 4.3. Distributive Networks

Recognising the relational worldview of Māori and the centrality of whānau, Kaupapa Māori theory highlights cultural structures that emphasise the collective as opposed to the individual [17]. Layered with various forms of cultural knowledge, practices, expectations and obligations, the concept of whānau sees relationships embedded within a culturally driven framework of co-operation, mutuality and reciprocity [27]. This provides a shared, collective support structure by which to mediate a range of challenges impacting on whānau wellbeing [18]. Integral to this is a reciprocal obligation on the part of individuals to invest in the wellbeing of the collective. Pre-existing connections, relationships and networks across iwi authorities, businesses, marae, community organisations, whānau, non-governmental organisations, and government agencies, served to swiftly unlock resources otherwise not available for collective community use [10,13]. The power of these distributive networks grounded within enduring, everyday cultural values and practices such as whanaungatanga (through which social relationships are embedded and understood) and manaakitanga (which determines the obligation to act in ways beneficial to the whole community) are emphasised [5,10,11].

## 5. Conclusions: Activating Mana Motuhake


*[The] response to COVID was one distinct moment in time. And that’s when we flourished. (*
*Rōpū Kaitiaki Wānanga, November 2020*
*)*


It is widely recognised that iwi, hapū and rōpū Māori excelled in their capacity to swiftly mobilise resources to where they were needed across a range of urban, rural and remote communities during COVID-19 [10,11,20]. Accepted as being “harder and faster” than the central government COVID-19 response [11] (p. 9), innovative Māori-led responses were fundamental to the pandemic response in Aotearoa [10,13]. That iwi, hapū and rōpū Māori were at the centre is of no surprise [5,10,13]. Māori have a long history of leading adaptive and innovative support during times of crisis in Aotearoa [5]. Anchored by an holistic worldview that prioritises relational cultural values, Māori-led COVID-19 responses drew on long developed and immediately available systems of culturally-embedded resources [15].

The significance of the outcomes achieved by Māori-led pandemic responses become further elevated when COVID-19 is understood as providing a unique opportunity for iwi, hapū and rōpū Māori to authentically activate foundational principles of a disruptive Kaupapa Māori theory. Approaches centred in Indigenous knowledge are ‘disruptive’ in that they seek to explicitly challenge the status quo so as to effect some form of radical change [28]. Deliberately intended to disturb “the sense of comfort and complacency that is characteristic of Western epistemologies” [29] (p. 3), when applied to a health system context, a disruptive approach will not only displace old systems, but also create new functionalities, particularly in relation to addressing the needs of those previously underserved [28].

Kaupapa Māori theory is explicitly concerned with structural transformation; critically analysing and understanding Western knowledge bases and power structures [17]. The urgency with which the COVID-19 crisis needed to be met meant the wider, dominant system was essentially forced out of the way. Of necessity, relationships were redefined, with iwi, hapū and rōpū Māori engaging in high-trust relationships with government agencies as they mobilised without hesitation to immediately deliver comprehensive responses. This relationship reset also saw proactive and agile, culturally-driven leadership fully, and unapologetically, activated.

Despite grave fears regarding the impact on COVID-19 on Māori, the COVID-19 pandemic has been described as the “only example in our contemporary history of the Māori community having better social outcomes than non-Māori” [10] (p. 135). The COVID-19 experience in Aotearoa demonstrated the outcomes able to be achieved when the wider system, long characterised by persistent underfunding for Māori providers, fragmentation, and competitive contracting models [4], functions to genuinely activate localised, self-determined, collective approaches. The success achieved by iwi, hapū and rōpū Māori exemplified what is possible when inflexible, slow and controlling government systems, incapable of responding quickly or effectively, are not only forced aside, but are replaced by self-determining Indigenous leadership, which delivers in ways government agencies are simply unable to [2,11,15]. No longer dictated to, or constrained in vision or aspiration, Māori-led responses that activated foundational principles of Kaupapa Māori theory, particularly in the early days of the COVID-19 pandemic in Aotearoa, were able to assume significant autonomy and control. This was both in relation to identifying and managing risks unique to their communities, alongside activating innovative and comprehensive, locally-led and driven responses, which positively benefited all within those communities.

Although the Māori-led COVID-19 response was largely positively received by the government and wider Aotearoa [10], the data from this study shows that once the crisis period ended, the government and its agencies returned to pre-pandemic patterns of control. Not only did initial high trust relationships disappear, but as seen in the findings from this study, there were attempts to downplay the significant role played by Māori, as well as change the overall narrative surrounding Māori-led responses.

The government contracting environment remains highly prescriptive, with Māori still severely constrained in their ability to deliver relational, whānau-centred, Te Ao Māori driven solutions [16]. Māori-led solutions continue to be positioned as an attachment to the wider, unchanged, dominant system, not only inhibited by narrow silos designed for and controlled by the wider system, but also accountable to dominant systems founded upon Western worldviews. As has been the call for many decades, system-wide changes focused on addressing institutional racism are necessary to realise aspirations for mana motuhake in health and wellbeing [4,12,30].

The success of Māori-led COVID-19 responses has created expectations moving into the future. As others have concluded, it is no longer acceptable to position iwi, hapū and rōpū Māori as mere stakeholders in decision-making, planning and policy see [10,12,13,15,20,31]. However, this post-crisis return to ‘business as usual’ by the government and its agencies indicates there is a long way to go before a genuine relationship reset can occur.

There has been much commentary focused on the opportunities for ‘re-set’, which COVID-19 has afforded Aotearoa. Māori-led COVID-19 responses have provided a chance to celebrate the ways in which communities looked after one another, as well as grow capacity to face future pandemics [5,13,14]. However, the unique opportunity for iwi, hapū and rōpū Māori to genuinely activate mana motuhake has afforded us significantly more than this. The success of COVID-19 Māori-led responses clearly illuminated not only the extent to which the system itself comprises a significant barrier to realising equitable outcomes for Māori, but more importantly, the vast possibilities able to be realised when transformative Kaupapa Māori theory is activated, and Indigenous solutions are enabled to be self-determining. In the face of a growing evidence base that removes any lingering doubts that mana motuhake lies at the heart of health and wellbeing for Māori, there must be a genuine commitment in Aotearoa to activate fundamental relationship resets [5,12,15]. To act otherwise is to ignore the now well evidenced proposition that mana motuhake provides a proven pathway to address inequities and enhance wellbeing not only for Māori but all in Aotearoa [10]. This is the great opportunity for reset, which must now be grasped.

## Data Availability

The data for this study is currently securely held by Te Hikuwai Rangahau Hauora | Health Services Research Centre, Te Herenga Waka–Victoria University of Wellington, at their premises in Wellington, Aotearoa. Individual participant data will not be available. This was a requirement of the ethics approval for confidentiality of information Victoria University of Wellington Human Ethics Committee—Application number 0000027171].

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
