# Peer review of "Enacting Mana Māori Motuhake during COVID-19 in Aotearoa (New Zealand): “We Weren’t Waiting to Be Told What to Do”"

_ijerph, 2023, doi:10.3390/ijerph20085581_

Round 1
Reviewer 1 Report
Thanks for the opportunity to review this work. The article's topic is fascinating because it shows us the culturally comprehensive response of indigenous communities to COVID-19.
methodology
I suggest basing on the COREQ guide or any other guide for the presentation of the methodology and results.
I suggest you comment on the interview process, the time spent on each interview, the type of analysis that was carried out, and whether the community was involved or only the researchers.
How was the reflexivity process for data analysis?
How was the data or sample saturation process carried out?
How did you build the codebook?
It would be convenient to know the type of questions asked by the researchers in the interview.
Regards
Author Response
- COREQ guide/types of questions asked/interview process/time spent on each interview/whether community was involved in interview process or only the researchers: Thank you for your suggestion to consider use of the Consolidated Criteria for Reporting Qualitative Studies (COREQ) in the reporting of our study’s interviews and wānanga (group discussion). We have accordingly expanded our methodology section in the paper to include: explanation of Kaupapa Māori theory (which prioritises Māori worldview, knowledge and analytical frames in the research) and its utilisation in the study’s data collection, as well as the initial use of Q methodology in the wānanga; reference to the professional, clinical and cultural guidance of the study by kaumātua (respected Elders) and a Rōpū Kaitiaki; and the undertaking of an initial literature review focused on Indigenous experiences of health and social services during pandemics or other global crises, conducted through an Internship. In expanding the information about the Kaupapa Māori theory-based approach to the interview process with the 27 leaders in Māori health, we have explained that a set of guiding questions sought to understand what oranga (wellbeing) looked like for Māori, what a Māori response to achieving oranga was, and what was required for services to operate effectively towards this goal. Further questioning investigated how Māori maintained oranga during the COVID-19 lockdown as individuals, and how their iwi, hapū, whānau and hapori (communities, including local services) did the same. All interviews were conducted by Māori interviewers and recorded with participants’ consent. Some were conducted in person, kanohi ki te kanohi (face-to-face), others over Zoom; the time spent varied. Irrespective of the method, all incorporated tikanga (Māori custom) and te reo Māori (the Māori language) in their process, which included understandings of Indigenous data sovereignty. This understanding ensured participants were offered the opportunity to review and amend the transcripts of their interviews and determine their identification alongside any quote/s used of theirs in reporting of study findings, which supported their continued ownership of their experiences and stories.
- Type of analysis carried out/reflexivity process for analysis/data or sample saturation/codebook building: At the conclusion of each interview, the researchers undertook a reflexive exercise through the discussion of their interpretations of the rich insights shared by the participants. As Indigenous researchers, this process allowed for acknowledgment of relationship with participants, whilst ensuring subjectivity did not influence research outcomes. Three leaders in Māori health who were initially identified as potential participants were unable to be interviewed. However, the data collected from those who were led to a natural point of data saturation needed to draw necessary conclusions. Transcribed interviews were iteratively analysed, the preliminary data initially being coded in NVivo, and the full dataset later being manually coded, using a systematic, thematic approach to aggregate and disaggregate the identified high-level and sub-themes. The analysis process undertaken by the researchers, classifying, sorting and arranging data, was time-consuming but needed for familiarisation with the data, the generation of codes, and construction and review of themes. The intricacies of this coding have not been included in the paper.
Reviewer 2 Report
This is a very interesting article that introduces to international scholars an important but lesser-known case. By looking at a group of local Māori leaders in New Zealand (Aotearoa), it shows the fascinating way in which communitarian responses to the COVID-19 pandemic were articulated. Largely based on qualitative research on 27 local health leaders, it explores the delivery of primary health care through the concept of mana motuhake (sometimes translated as "the right or actual practice of self-government”, here rendered as “autonomy, self-determination, independence, sovereignty, authority and control over one’s own destiny “). Then glossary is very useful.
The article, however, moves in a kind of theoretical and conceptual vacuum. It remains incomplete insofar as it is deprived of a larger broader framework in which to contextualize the core argument. I could suggest various notions and conceptual paths, but in this commentary, I will limit myself to one or two.
A good starting point would be, to begin with exploring the notion of "exemplary ethical communities" and see how this concept can fit the cause and argument about Māori leadership into a more internationally understandable framework.
EEC are a particularly suitable framework when dealing with self-determining, collective, Indigenous leadership.
Also important would be to show the broader implications of the interconnectedness of contemporary human crises -- even though it my appear to be outside the scope of the article: The pandemics should be understood and placed within the broader context of biodiversity loss and climate change, of which traditional leaders have repeatedly expressed more awareness than political leaders.
As the article describes how this success story was achieved for "everyone in Aotearoa, the additional notion of "exemplary nation-states "could be added to its argumentative and explicative power in a broader international context. Perhaps it could be said that this is where two forms of exemplarity have joined forces: even if "the wider system was forced to step aside", it was the entire public system which benefited from, and had to gain from, the impact of Māori leaders actions.
Finally, I also suggest changing the title by clarifying that it is about Māori.
Once these conceptual revisions are introduced, I think the article can be published and become more clearly understandable beyond the given case study indicated.
Author Response
- Include broader theoretical framework in which to conceptualise core argument/exemplary nation-states could be added to its argumentative and explicative power in a broader international context: The theoretical frameworks suggested (i.e., exemplary ethical communities/ exemplary nation states) were reviewed. However, Kaupapa Māori theory was considered the most appropriate theoretical framework in which to position the core argument of this article. Kaupapa Māori theory is internationally known across Indigenous communities. Please see: See 76-95; 109-116; 589-595; 633-636; 663-679; 697-714; 726-732; 760-771
- Show the broader implications of the interconnectedness of contemporary human crises - even though it my [sic] appear to be outside the scope of the article: We agree with this reviewer that addressing this point is outside the specific scope and focus of this article.
Reviewer 3 Report
This is a well-researched and important paper with a marked underlying political agenda.
There is one rather irritating feature of the text. Whilst there is a full glossary of Maori terminology at the end of the paper, the integration of these terms into the text without the English meaning being given in brackets on first use is not helpful. The reader has repeatedly to look up each term as they proceed. Thus, as an example, 'readers beyond Aotearoa (New Zealand)' might well respond more positively if this textual concession were made.
Author Response
- Integrate English translations of Te Reo Māori into the article: At the first use of a Māori word in the paper, an English language meaning now follows in parentheses. If a Māori term is used that cannot easily be explained, the meaning is included instead in a footnote. The Glossary of Māori words has also been expanded.
Round 2
Reviewer 1 Report
Dear Editor
The topic is very interesting and the suggested changes are correct.
Regards